# Land Use-Driven Changes in Ecosystem Service Values and Simulation of Future Scenarios: A Case Study of the Qinghai–Tibet Plateau

**Yongkang Zhou [1], Xiaoyao Zhang [2], Hu Yu [3],\*, Qingqing Liu [4] and Linlin Xu [3]**

1   School of Economics, Guizhou University, Guiyang 550025, China; Zhouyk2020@163.com
2   School of Geography and Tourism, Anhui Normal University, Wuhu 241000, China; zxy9797297@163.com
3   Institute of Geographic Sciences and Natural Resources Research, CAS, Beijing 100101, China; xulinlin20@mails.ucas.ac.cn
4   College of Tourism and Exhibition, Henan University of Economics and Law, Zhengzhou 450046, China; qingqing_liu@huel.edu.cn
\*   Correspondence: yuhu@igsnrr.ac.cn; Tel.: +86-010-64889216

**Abstract:** Global climate change and land use change arising from human activities affect the ecosystem service values (ESVs). Such impacts have increasingly become significant, especially in the Qinghai–Tibet Plateau (QTP). Major factors impeding the construction of China's "ecological security barrier" are shifts in land-use patterns under rapid urbanization, irrational crop and animal husbandry activities, and tourism. In the present study, land use changes in the QTP in recent years were analyzed to determine their impacts on ESVs, followed by simulations of the interactive and evolutionary relationships between land use and ESVs under two scenarios: natural development scenarios and ecological protection scenarios. According to the results, the QTP land-use structure has a small change, and the main land use type is alpine grassland, followed by bare land and woodland. The stability of the major land use types is the key factor responsible for the overall increasing ESV trend. Different regions on the QTP had substantially varied ESVs. The northwest and southeast regions are mostly bare land, which is a concentrated area of low value of ecosystem services. A variety of land use types including grassland and woodland have been found in the humid and semi-humid areas of the central region, so the high value of ecosystem services is concentrated in this area to form a hot spot, with a Z value of 0.63–2.84. Simulations under the natural development and ecological protection scenarios revealed that land use changes guided by ecological policies were more balanced and the associated ESVs were relatively higher than those under the natural development scenario. Under a global climate change context, human activities on the QTP should be better managed. Sustainable development in the region could be facilitated by ensuring synchronization between resource availability and adopted socioeconomic activities.

**Keywords:** land use changes; ecosystem service values; Qinghai–Tibet Plateau

## 1. Introduction

The Sustainable Development Goals (SDG 2030) include the establishment of good human–nature relationships and the enhancement of ecosystem service (ES) provision, to ensure the sustenance of economies and livelihoods of communities effectively. Land use, which is a major link between ESs and human activities [1], facilitates the achievement of diverse needs in human society. However, land use activities could adversely affect ecosystem structure and processes, and drive the evolution of such processes and functions [2]. From a global socioeconomic development perspective and ES evolution, land use structures influence the levels and threshold ranges of ESs that have benefits [3], through the modification of factors such as ground vegetation and landscape characteristics. Today, intense and extensive human activities—including globalization, regional urbanization,

local industrialization, agriculture, animal husbandry, and tourism—have had considerable impacts on regional and local environments. Such activities constantly alter land surface structure and influence the relationships among components of natural ecosystems, spatial distribution of resources, as well as material circulation. Consequently, different regions are affected by different factors and their interaction to different degrees, so the evolution of ES values (ESVs) shows big differentiation in different natural ecological types of areas, which is necessary to optimize and regulate from the policy level.

The Qinghai–Tibet Plateau (QTP) is not only the largest plateau in China, but is also the plateau located at the highest altitude globally. The plateau has been subjected to diverse land use activities over the years, ranging from nomadic activities to farming, with a modern tourism economy being the current mainstay. In addition, the urban patterns in the region have shifted from a central city system to a system of regional cities and towns, and the land use structure is increasingly diversifying and becoming larger in scale [4]. During such transition processes, land use change associated with human activities have more obvious impacts on the ESVs of the QTP, and are mainly manifested as follows: (i) destruction of the grassland ecosystem caused by unregulated livestock rearing and conflict between food production and livestock rearing, (ii) uncontrolled infrastructure development to support tourism development, and (iii) high volumes of industrial migrants regionally above the land carrying capacity. The QTP is an important ecological security barrier in western China [5], and the foundation of ESVs for development in the region is being affected adversely by changes in the land use structure arising from human activities, resulting in challenges such as land shortage and human–nature conflicts.

The present study investigated the coupling relationship between shifts in the QTP land use structure and its ESVs. The aims of the present study were to explore (i) changes in the QTP's land use structure, (ii) spatial trends of ESVs under land use change, and (iii) future trends under natural evolution and ecological protection scenarios. The findings of the present study could facilitate the formulation of policies to support the sustainable exploitation of ESs in the QTP.

### 1.1. Land Use Change and Ecosystem Services

Land use change is a major aspect via which human activities influence ecosystem services and function. The subsequent changes in surface vegetation structure and patterns have major impacts on ecosystem structure and function. In addition, changes in land use type and management methods also influence ESs. For example, overgrazing modifies the characteristics and scale of the land cover, which, in turn, affect primary productivity, habitat and species diversity, in addition to biogeochemical cycles and energy flow patterns [6].

Numerous studies have investigated the factors affecting ESs. Research on ES emerged in the 1970s [7], and has attracted the attention of diverse scholars over the years [8]. In 2005, the United Nations clarified the relationship between ecosystems and land use in the Millennium Ecosystem Assessment (MA) and classified the various types of ESVs [9]. Through the examination of land use types and their change, in addition to shifting land use patterns in space [10], scholars have been able to analyze the factors driving the establishment of different ESs.

### 1.2. Evaluation and Simulation of Ecosystem Service Values

ESVs are dynamic but predictable. The development of remote sensing and geographical information systems technologies has facilitated the emergence of various ES classification and evaluation systems worldwide, as well as a batch of ecological models, such as Integrated Valuation of Ecosystem Services and Tradeoffs, Artificial Intelligence Ecosystem Services, Revised Universal Soil Loss Equation, Soil and Water Assessment Tool, Cellular Automata (CA), and gray forecasting models [11]. Such models have been applied extensively in quantitative research on the supply of single or multiple ESs at various scales, ranging from local to global scales [12]. Some scholars have further achieved the

monetization of ESVs by estimating the per-unit pricing of ES functions or using equivalent value per unit area of ES as the basis [13]. Such models and tools have provided new opportunities and approaches for the evaluation and prediction of ESVs.

Although there are numerous studies that have explored the relationship between land use and ESVs, simulation studies based on historical data exploring potential future trends in the relationship between land use and ESVs under multiple scenarios are largely lacking. The CA-Markov model combines the capacity of Markov Chain Monte Carlo methods to predict long-term series and the advantages of CA in simulating spatial changes in complex systems. The model is significantly superior to other physiological and ecological models in terms of goodness-of-fit, simulation dynamics, and multi-scenario land use analysis [14]. For example, Sun et al. used the CA-Logistic-Markov model to analyze changes in land use patterns and the spatial distribution of ES regionally in the Atlanta metropolitan area in 1985–2012. They also simulated ES characteristics under four landscape scenarios in 2030 to explore the optimal land use strategies that could enhance ESs [15]. The quantitative characteristics and spatial relationships of ES supply and demand were determined by matching the spatial layout of the supply and demand. Research carried out in the QTP has seldom involved the quantification of ES demand, and has rarely paid attention to the ES characteristics changes over the long-term. To better grasp future macro trends and facilitate the formulation of relevant policies, the CA-Markov model can be used by setting up separate scenarios of natural development and ecological protection, to simulate future land use change and their impacts on ES.

### 1.3. Land Use Change on the Qinghai–Tibet Plateau

The QTP is a unique geographical feature and the highest plateau in the world, with its average altitude being greater than 4000 m. It is a critical ecological security barrier for China and has extremely high ESVs associated with water conservation and hydrological regulation [16]. In recent decades, the QTP has been experiencing the Westerlies and the East Asian and Indian monsoons at increased intensities, which could have altered the distribution of water vapor in the region [17]. In addition, the average temperature in the region has increased substantially since 1997, and significantly after 2005 [18]. Such shifting environmental conditions have introduced climatic differences between the southeastern and northwestern regions of the QTP, with generally warmer and humid conditions [19]. Compounded by various factors, including the sustained intensification of human activities, grassland degradation, desertification, expansion of urban spaces, glacial melting, and particularly, dynamic shifts in alpine grassland vegetation communities, ESVs have been affected considerably.

The QTP is highly vulnerable to the effects of climate change and human activities, so that the regional environment is both climate-sensitive and ecologically vulnerable. In addition, the region is an important base in China for animal husbandry due to the grassland resources, with marked increases in grazing activities and population [20]. Demand by the Chinese population for high-quality livestock products (milk, meat, and fur goods) from naturally-fed animals, it has been driving the increase in the quantity and intensity of grazing in the region annually. Due to the considerable changes in the regional climate and environment in QTP in recent years, coupled with climate warming, water pollution, grassland degradation, and desertification, sustainable socioeconomic development at the local level has been seriously impaired [21].

Studies have reported that changes in QTP's ES capacities in 1995–2015 were either stable or rising [22]. An increase in ES demand in the eastern region of the QTP was especially obvious, which could have been due to the impacts of climate warming and human activities on the QTP ecosystem. Notably, ES supply decreased significantly in areas that had undergone land use and structure change associated with marked permafrost degradation, and land desertification or natural disasters [23]. In addition, the overall ES demand has increased continuously in areas with widespread human activities, bringing

about greater challenges with regard to the sustainable supply of ES services regionally, as well as their regulation functions [24].

Considering the emerging challenges above, there is a need to explore the mechanisms via which land use change affects ESs on the QTP, in addition to the interactions between the two factors. Such studies could provide an essential reference resource and facilitate the formulation of appropriate ecological protection measures and sustainable regional development in the QTP. The present study used land use data spanning a 23-year period and ESV accounting methods to estimate ESVs in the QTP in 1992–2015. In addition, we examined the spatiotemporal distributions of the estimated ESVs. With the historical patterns of ESV changes as the basis, the CA-Markov model was used to predict the ESV trends in the QTP's ESVs in 2035 under two scenarios: natural development and ecological protection, in turn, revealing the relationship between land use and ESV on QTP.

## 2. Materials and Methods

### 2.1. Study Area

The QTP (26°00′–39°47′ N, 73°19′–104°47′ E) is located in central Asia and occupies an area of approximately $250 \times 10^4$ km$^2$ within China's territories. It accounts for approximately one quarter of the country's total land area, encompassing the Tibet and Xinjiang Uygur Autonomous Regions, the entire Qinghai province, and parts of the Tibetan areas in the Sichuan, Yunnan, and Gansu provinces. The QTP is an important component of China's strategic layout for ecological security [25], which includes the "two screens and three belts." It is also the "riverhead" and "ecological source" of numerous rivers in China and South and Southeast Asia, including the Yangtze, Yellow, Lancang, and Yarlung Tsangpo Rivers (Figure 1), and provided diverse ESs such as water and soil conservation and biodiversity maintenance for China and its surrounding countries.

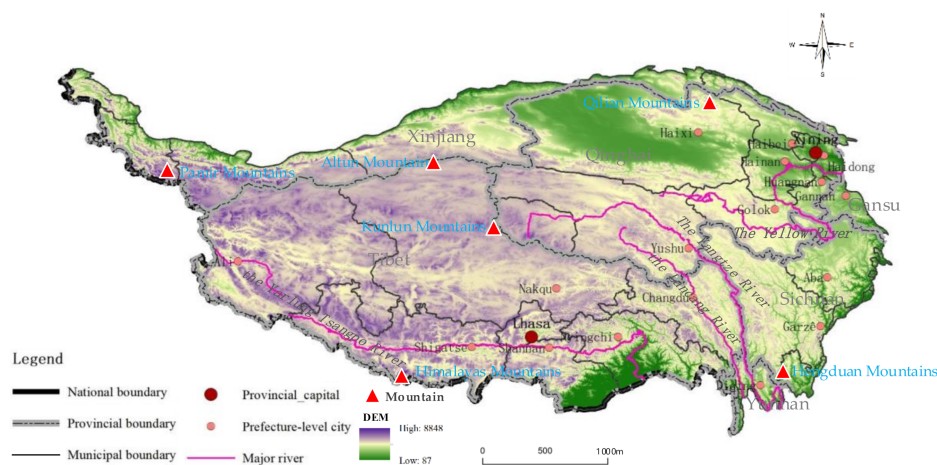

**Figure 1.** Study area overview.

The QTP is composed of a series of mountain ranges, which include the Hengduan, Qilian, Kunlun, Altun, Himalayas, and Pamir Mountains, from east to west. The regional climatic characteristics, including severe cold, drought, and oxygen deficiency, render the QTP's ecological environment extremely fragile and sensitive, with poor self-regulation and resilience. In modern times, human activities in the QTP have included large-scale water and mineral resource exploration. In addition, immigration, accelerated industrialization and urbanization, increased transportation, and unregulated crop and animal husbandry activities have become widespread. Such activities have led to the continuous expansion of urban built-up areas and severe damage to grasslands. In 1990–2010 alone, human activity intensity on the QTP increased by approximately 30% [26].

### 2.2. Data Sources

Spatial data of the administrative regions, land use data, socioeconomic development statistics, and natural and geographic environment data were used as explanatory variables in the present study. The respective sources were as follows: (i) spatial data on administrative regions were acquired from the official website of the State Bureau of Surveying and Mapping; (ii) land use data at a resolution of $300 \times 300$ m$^2$ for three periods of 1992, 2005, and 2015, were obtained from the National Tibetan Plateau Data Center (http://data.tpdc.ac.cn/zh-hans/, accessed on 29 April 2020). The data were mainly based on the 2015 ESA GlobCover data. The conversion rules of the QTP land cover classification system were constructed, the confidence function of land cover classification and land type fusion rules were constructed, and the land cover product fusion and modification were carried out. The production of this land use data has gone through five stages: remote sensing image data preparation, classification system establishment, interpretation mark establishment, classification interpretation stage, and accuracy inspection of interpretation results. The accuracy inspection mainly relies on high score images and field empirical combination, so the Kappa value of the land use data is significantly higher than 90%; (iii) socioeconomic development statistics (equivalent factors for value enhancement), including grain output per unit area, total crop income, and national grain output per unit area of the various QTP counties, were primarily obtained from the *China Statistical Yearbook 2016,* and statistical yearbooks of the respective provinces and cities; (iv) geomorphological data, including elevation, topography, and surface water characteristics (rivers and lakes), were downloaded from the Geospatial Data Clouds. After exploring relevant literature [27], seven explanatory variables were selected from the natural and socioeconomic dimensions, including altitude, slope, population per unit area, night-time light data, and respective distances from rivers/lakes, roads, and the city center.

### 2.3. Matrix for Land Use Transfer

This matrix mainly reflects the dynamic process of the area transformed between the initial and final land use types in the QTP over the study period. The equation is as follows:

$$Sij = \begin{bmatrix} S_{11} & S_{12} & \cdots & S_{1n} \\ S_{21} & S_{22} & \cdots & S_{23} \\ \cdots & \cdots & \cdots & \cdots \\ S_{n1} & S_{n2} & \cdots & S_{nn} \end{bmatrix} \tag{1}$$

where $S$ is the area, $n$ is the number of land use types, $i, j = (i, j = 1, 2, \cdots, n)$ are the represented land use types before and after transformation, and $Sij$ is the area transformed between land use types $i$ and $j$.

### 2.4. Ecosystem Service Value Estimation and Coefficient Correction

Ecosystems in China are typically complex and diverse, they include boreal coniferous forest ecosystems in cold temperate zones, mixed coniferous and broad-leaved forest ecosystems in middle temperate zones, deciduous broad-leaved forest ecosystems in warm temperate zones, subtropical evergreen and deciduous broad-leaved mixed forest ecosystems, etc. On the Qinghai-Tibet Plateau, forest, shrub, alpine grassland, alpine meadow, and alpine desert ecosystems are distributed from southeast to northwest, with lakes and marshes intersecting with each other, which constitute the rich ecosystem diversity of the Plateau. In the present study, the equivalent factor for value per unit area was applied (Table 1). Ecosystem services are divided into nine categories: food production, raw material production, gas regulation, climate regulation, hydrology conservation, waste treatment, soil formation and protection, biodiversity maintenance, and leisure and entertainment. Equivalence ecosystem service value refers to the potential capacity of an ecosystem to produce relative contribution of ecological services, which is defined as the economic value of the annual natural grain production of farmland with the national

average yield of 1 hm$^2$. Based on this, the weighted factor table can be converted into the unit price table of ecosystem services in that year [28]. The method was developed by Xie et al. on the basis of the research by Costanza et al. and states that the equivalent factor for ESs is equal to 1/7th the value of grain per unit area. The equations for estimating the ESVs are as follows:

$$ESV_f = \sum \left( A_k \times VC_{fk} \right) \tag{2}$$

$$ESV = \sum_{k=1}^{n} \left( A_k \times VC_k \right) \tag{3}$$

where $ESV_f$ is the functional value of the $f$th ecosystem service, $VC_{fk}$ is the functional value coefficient of the $f$th land use type $k$, $ESV$ is the total ESV, and $A_k$ is the area under land use type $k$.

**Table 1.** Equivalent ecosystem service value per unit area.

| Ecosystem Classification | | Supply Services | | Regulating Services | | | | Support Services | | Cultural Services |
|---|---|---|---|---|---|---|---|---|---|---|
| Primary | Secondary | Production | | Regulation | | | Waste Disposal | Conservation | | Aesthetic Landscape |
| | | Food | Raw Material | Gas | Climate | Hydrology | | Soil | Bio-Diversity | |
| Farmland | Dry land | 0.85 | 0.40 | 0.67 | 0.36 | 0.27 | 0.10 | 1.03 | 0.13 | 0.06 |
| | Paddy field | 1.36 | 0.09 | 1.11 | 0.57 | 2.72 | 0.17 | 0.01 | 0.21 | 0.09 |
| Woodland | Coniferous | 0.22 | 0.52 | 1.70 | 5.07 | 3.34 | 1.49 | 2.06 | 1.88 | 0.82 |
| | Coniferous mixed | 0.31 | 0.71 | 2.35 | 7.03 | 3.51 | 1.99 | 2.86 | 2.60 | 1.14 |
| | Broadleaf | 0.29 | 0.66 | 2.17 | 6.50 | 4.74 | 1.93 | 2.65 | 2.41 | 1.06 |
| | Bush | 0.19 | 0.43 | 1.41 | 4.23 | 3.35 | 1.28 | 1.72 | 1.57 | 0.69 |
| Grassland | Grassland | 0.10 | 0.14 | 0.51 | 1.34 | 0.98 | 0.44 | 0.62 | 0.56 | 0.25 |
| | Shrubland | 0.38 | 0.56 | 1.97 | 5.21 | 3.82 | 1.72 | 2.40 | 2.18 | 0.96 |
| | Meadow | 0.22 | 0.33 | 1.14 | 3.02 | 2.21 | 1.00 | 1.39 | 1.27 | 0.56 |
| Wetlands | Wetlands | −0.51 | 0.50 | 1.90 | 3.60 | 24.23 | 3.60 | 2.31 | 7.87 | 4.73 |
| Desert | Desert | 0.01 | 0.03 | 0.11 | 0.10 | 0.21 | 0.31 | 0.31 | 0.12 | 0.05 |
| | Bare ground | 0.00 | 0.00 | 0.02 | 0.00 | 0.03 | 0.01 | 0.02 | 0.02 | 0.01 |
| Water | Waterbody | 0.80 | 0.23 | 0.77 | 2.29 | 102.24 | 5.55 | 0.93 | 2.55 | 1.89 |
| | Glacier Snow | 0.00 | 0.00 | 0.18 | 0.54 | 7.13 | 0.16 | 0.00 | 0.01 | 0.09 |

*2.5. Prediction of Future Scenarios*

In this paper, the natural development scenario is constructed based on the assumption that the rate of land use conversion in the historical period remains unchanged [29], and the influence trend of various suitability factors on land use change will continue to be consistent with the current situation. Under the natural development scenario, the rules of transfer between various land use types in 1992–2015 were adopted as the baseline scenario for forecasting future land use change. Under the ecological protection scenario, areas within the QTP's main functional regions that are prohibited from development were integrated with all the nature reserves, national parks, geological parks, and scenic spots being demarcated as zones in which development is restricted. These were combined with

urban land and agricultural land expansion to comprehensively study and evaluate future development scenarios. The natural development scenario can be regarded as the benchmark scenario for future land use changes in the Qinghai-Tibet Plateau. The ecological protection scenario is based on the natural development scenario, giving priority to the protection of basic farmland, ecological protection areas, etc., and relaxing restrictions on urban development on the premise of respecting the natural ecosystem and reasonable environmental carrying capacity. This scene follows the principles of sustainable development and emphasizes the coordinated development of protection and development [30]. The ecological protection scenario is a comprehensive and sustainable future land use change scenario and an ideal scenario for future land use changes on the Qinghai-Tibet Plateau.

The CA model is a kinetic model that is spatiotemporally discontinuous and in which time, space, and state are discrete. The model is mainly composed of cells and their states, cell spaces, cell neighborhoods, and conversion rules. Each cell within the cell space is at its self-delimited state, and is updated synchronously according to its self-defined local rules. When the local rules interact, a dynamic evolutionary system is established. A different approach is the Markov model. It is a method of predicting the probability of an event based on Markov's theory, and is mostly considered a process of predicting future land use trends [31]. Since CA focuses on simulating spatial changes in complex systems, whereas Markov predicts future land use change trends, the CA-Markov integrated model combines the advantages of both, allowing simulations of land use change to concurrently account for spatial and temporal changes. The specific simulation process of the CA-Markov model is illustrated in Figure 2.

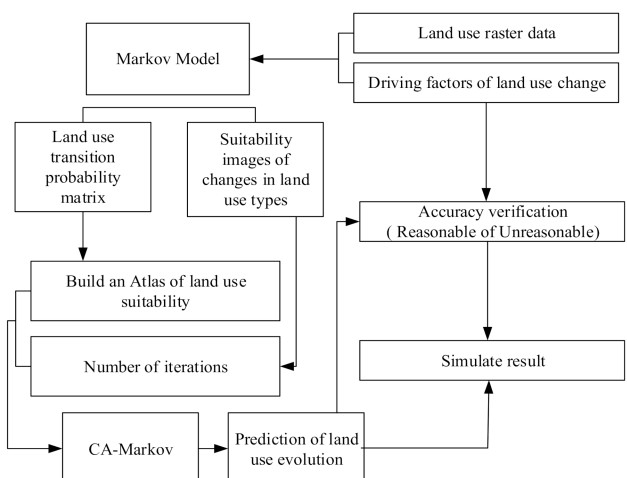

**Figure 2.** CA-Markov model simulation process.

We performed the following method action. Firstly, the cell size in the present study was set to $1 \times 1$ km$^2$ to accommodate the QTP's vast area. When one cell contained more than one land use type, the type with the largest proportion was considered the major type. The state of each cell corresponded to one of the following nine land use types: farmland, woodland, grassland, shrub land, urban land, waterbody, desert, bare land, and glacier. Then, a $5 \times 5$ km$^2$ matrix space surrounding each cell was used to analyze the cause(s) of significant changes in cell states to estimate the transformation matrix. The QTP's land use data in 1992, 2005, and 2015 were separately entered into the Markov model, the interval years were set to 10 years and 20 years, and the error ratio was set to 0.15 (indicating that the simulation accuracy was above 0.85). The respective transformation probability and area matrices of the land use structure in 1992–2005 and 1992–2015 were subsequently obtained. The IDRISI [32] multi-standard evaluation module was subsequently used to normalize the explanatory variables according to the research purposes, and the weighted linear combination calculation method was used to compare

the relative importance of the various driving factors. Based on the requirements under the corresponding scenario, the results were subsequently used to generate the transfer suitability diagram, which then served to set the number of iterations. 2005 was adopted as the starting year for the prediction of the spatial shifts in land use in 2015, with 10 years set as the number of iterations. The simulated land use data for 2015 were verified against the actual land use data in the same year to ensure the accuracy and reliability of the CA-Markov model's simulation results. Afterward, 2015 was used as the starting year, with the number of iterations set to 20 years, for the simulation of land use patterns in 2035. Finally, CROSSTAB, the Kappa coefficient verification method provided in IDRISI, was used to verify the simulation results and ensure their accuracy and reliability. In order to measure the consistency between the simulation results and the actual land use data, this study uses the Kappa value to evaluate the reliability of the model. The accuracy verification shows that the Kappa value is 0.9216, and the Kappa value > 0.85, indicating that the model has high simulation accuracy, meets the research needs, and can be used to predict future land use patterns.

## 3. Results

### 3.1. Land Use Change Characteristics on the QTP

According to the comparison table of "Three Categories" in "Land Use Status Classification" and "Land Management Law of the People's Republic of China", the land cover classification system of the QTP was constructed, and the land cover types in the QTP were divided into nine categories: farmland land, urban land, desert, bareland, grassland, forest, shrub, water, and glacier. Local adjustments occurred in QTP land use structure over the 23-year study period, with the two prominent changes being an increase in urban land and desert areas, and a decrease in shrubland and bare land (Figure 3). The QTP was predominantly grassland in 1992–2015, with the land use type accounting for 67.91% of the total area. This was followed by bare land and woodland, accounting for 14.59% and 9.01% of the total area, respectively. The three land use types jointly comprised 91.51% of the total area, with no obvious change in their predominance over time. Driven by urbanization and expansion of urban construction, the growth rate of urban land area reached 96.92%. There was also an obvious increase in desert area (17.31%) due to temperature, precipitation, and climate warming factors. Correspondingly, there were reductions in shrub land and bare land areas, at −4.41% and −9.42%, respectively. In addition, glacial melting decreased by 2.84%.

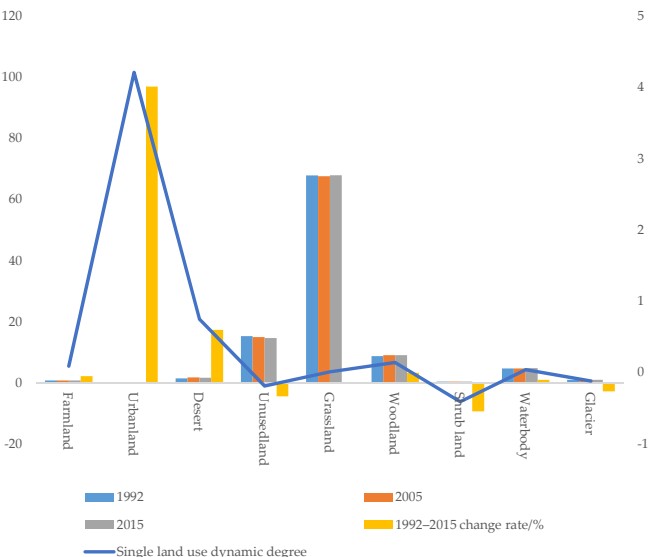

**Figure 3.** Changes and dynamics of land use area from 1992–2015.

Based on a land use dynamic change perspective, there were increases in urban land (4.21%), desert (0.75%), and woodland (0.14%), and decreases in bare land (0.19%), shrubland (0.41%), and glaciers (0.12%). Over the study period, overall land use change on the QTP was affected by both human activities directly and the indirect effects of the activities. The impacts of urbanization and global warming resulting from population agglomeration on the plateau were significant, while ecological environmental protection policies played key regulating roles. For example, in 1992–2015, grassland area decreased from 67.81% to 67.57%, which was as a result of policies that restricted grazing and promoted a reasonable balance in the grassland–livestock relationship. With the restriction on overgrazing, grassland area increased substantially.

### 3.2. Direction of Transfer of Land Use Types

From 1992 to 2015, the transfer activities of land use types on the QTP are multi-directional. Urban, desert, and shrubland accounted for relatively high proportions of the transformed areas. Land uses that were transformed included farmland, grassland, and waterbody, accounting for 93.09% of the total amount of transformed urban land (Table 2). On the whole, there is a significant mutual conversion phenomenon between grassland and desert. About 5093 km$^2$ of desert area is converted into grassland, and in other places, the amount of grassland transferred to desert is 6937 km$^2$. Shrubland was mainly transformed into woodland, accounting for 86.51% of the total area transformed. Glacier transformation into a waterbody was also widespread.

**Table 2.** 1992–2015 Qinghai-Tibet Plateau land use transfer matrix.

| km$^2$ | | 1992 | | | | | | | | | Land Area Gains and Losses |
|---|---|---|---|---|---|---|---|---|---|---|---|
| | | Farm Land | Wood Land | Grass Land | Shrub Land | Water Body | Urban Land | Desert | Bare Land | Glacier | |
| 2015 | Farmland | 16,503 | 335 | 1585 | 6 | 48 | 78 | 8 | 38 | 0 | 394 |
| | Woodland | 295 | 225,542 | 8781 | 1334 | 56 | 3 | 0 | 1 | 1 | 7493 |
| | Grassland | 973 | 2251 | 1,733,812 | 102 | 4789 | 88 | 5093 | 31933 | 43 | 2572 |
| | Shrubland | 1 | 216 | 140 | 8096 | 42 | 1 | 153 | 81 | 0 | −908 |
| | Waterbody | 120 | 129 | 6937 | 71 | 115,581 | 90 | 6 | 0 | 1367 | 1204 |
| | Urban land | 309 | 37 | 674 | 3 | 89 | 928 | 25 | 304 | 0 | 1166 |
| | Desert | 1 | 2 | 6935 | 6 | 80 | 2 | 31,236 | 5310 | 1 | 6431 |
| | Bareland | 5 | 7 | 17,610 | 20 | 1760 | 13 | 621 | 362,128 | 26 | −17,627 |
| | Glacier | 0 | 1 | 38 | 0 | 652 | 0 | 0 | 22 | 24,085 | −725 |

The land areas transformed into urban land, grassland, farmland, and bare land were quite high. The area of urban land on the QTP doubled over the 23-year period, accounting for 60.83% of the total transformed land use. At the same time, a portion of grassland and bare land had degenerated into deserts, with their areas accounting for 56.21% and 43.04% of the transformed area, respectively. According to the QTP land use transfer matrix, the transformation of bare land and shrubland into grassland and woodland, respectively, was particularly notable.

### 3.3. Ecosystem Service Values on the Qinghai-Tibet Plateau and Their Trends

Overall, grassland, waterbody, and woodland land use types were the major contributors to the ESVs. The combined contributions of the three land use types above were 98.55%, 98.55%, and 98.59% over the three years, successively (Figure 4). The major factor influencing ESV tends was land use type transfer. Although land use change affects the ecosystem value of different land types, it does not have a great impact on the whole.

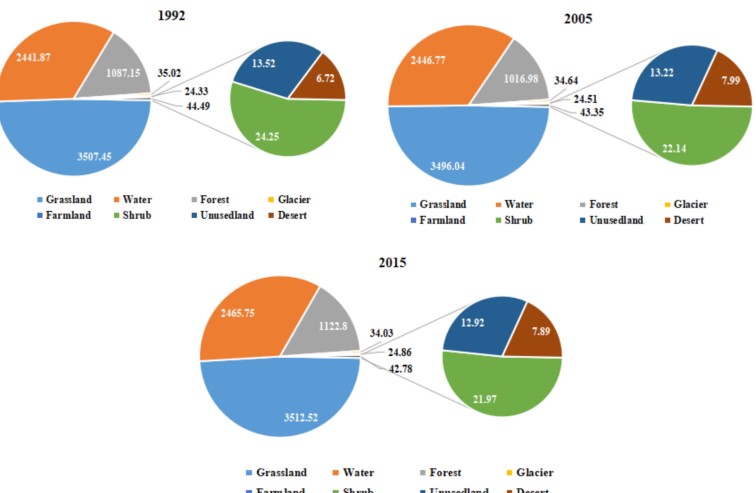

**Figure 4.** Ecosystem service values of the different land use types on the Qinghai-Tibet Plateau in 1992, 2005, and 2015.

From a dynamic change perspective, the ESV trends in the QTP exhibited decreases followed by increases (Table 3). The ESV was 7140.31 B in 1992, reduced to 7062.29 B in 2005, and then increased to 7202.74 B in 2015. The increase in the latter stage was greater than the decrease in the earlier stage, which was consistent with the overall trend of land use structure. The changes in the value in the two periods were −7.802 and 14.045 B yuan, respectively.

**Table 3.** Changes in ecosystem service values in various ecosystems in the Qinghai-Tibet Plateau from 1992 to 2015.

| $10^8$ yuan/hm$^2$ | ESV Change | | |
|---|---|---|---|
| | 1992–2005 | 2005–2015 | 1992–2015 |
| Farmland | 0.18 | 0.35 | 0.53 |
| Desert | 1.27 | −0.1 | 1.17 |
| Bare land | −0.3 | −0.3 | −0.6 |
| Grassland | −11.41 | 16.48 | 5.07 |
| Woodland | −70.17 | 105.82 | 35.65 |
| Shrubland | −2.11 | −0.17 | −2.28 |
| Waterbody | 4.9 | 18.98 | 23.88 |
| Glacier | −0.38 | −0.61 | −0.99 |
| Total | −78.02 | 140.45 | 62.43 |

Grassland had the highest ecological value, accounting for 48.77% of the total value. The value of deserts was the lowest, accounting for only 0.11% of the total value. The ESV produced per unit of waterbody was relatively high. Although the area of the waterbody land use type was only half that of woodland, its total ESV was higher than that of the woodland. The ESVs of bare land and glaciers declined continuously, with their magnitudes being greater in 2005–2015 than in 1992–2005. The main reasons were the continuous transfer of bare land to grassland and urban land, and conversion of glaciers to waterbodies, especially in the latter ten years of the study period, the influence of the external environment becomes more and more intense.

*3.4. Characteristics of Different Dimensions of Ecosystem Service Values and Their Changes*

3.4.1. Characteristics of ESVs of Different Eco-Geographical Areas

An eco-geographic area reflects the natural environment characteristics, so that the land use patterns derived would be similar in different sites with similar eco-geographic

features. Hence, the analysis of zonal characteristics of the ESVs in the QTP was analyzed from an eco-geographic area perspective. First, using the QTP's ESV scores as the basis, the ArcGIS natural breaks classification method was used to divide the plateau into five types of zones based on value: very low (VL), low (L), average (A), high (H), and very high (VH), which shows that the areas around the eastern Kunlun Mountains, Altun Mountains, and Qaidam Basin are mostly desert and bare land, which account for 17.38% and 0.95% of the total land use, respectively. They have low ESVs and belong to the VL-value zone. There are no large urban built-up areas in the areas, and not much change has occurred to the land use structure. The Sichuan–Tibet alpine-valley, located to the southeast of the QTP, is an A-value zone and accounts for only 9.01% of the total value. H- and VH-value zones, which are mainly distributed in the Qilian Mountains, QTP, and valleys in southern Tibet, account for 72.66% of the total value (Figure 5). The VL-value zones in northern QTP had a decreasing trend over the study period, with the reductions in the Altun Mountain–Qaidam Basin belt being the most prominent. The ESV changes in the belt were mostly from the L- and VL-value zones to the A- and H-value zones. ESV increases were more notable in the belt along the Gangdise Mountain and the edge of the Himalayas and were mainly from the A- and H-value zones to the VH-value zones.

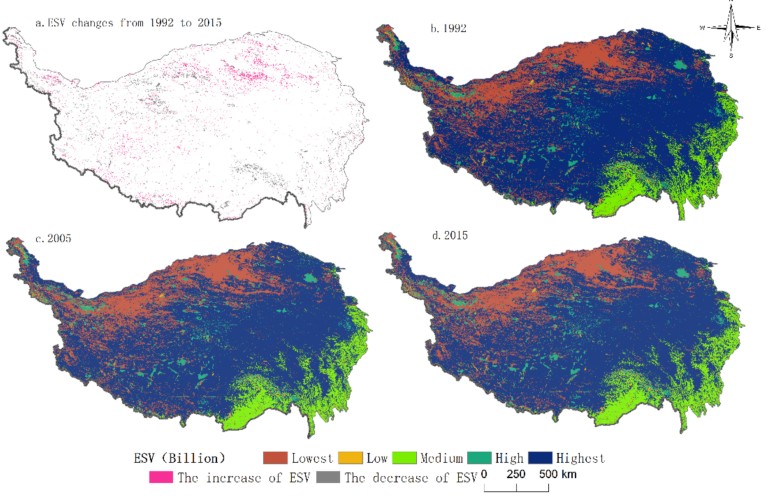

**Figure 5.** Overall evolution characteristics of the ecosystem service values of the Qinghai-Tibet Plateau from 1992 to 2015.

From 1992 to 2015, the area where the ESV on the QTP decreased accounted for 3.17% of the total area, and the area of decreased value was mainly concentrated in the southern area of Kunlun Mountains - Altun Mountains and the eastern part of the southern valley of Tibet. The main reason for the decrease in the ESV in the two regions is that the aquatic and grass conditions are relatively good and suitable for grazing activities. Disorderly human grazing activities aggravate the inherent vulnerability of the region and ultimately lead to the deterioration of the regional ecological environment. In different regions, the trend of ESVs in the QTP is different, with some places increasing and some places decreasing. However, on the whole, the decreasing area of ecosystem service value is less than the increasing area of ESVs. This suggests that the implementation of effective national grazing management and farming strategies adapted to local conditions has optimized the ecological environment of the QTP.

### 3.4.2. Variations in ESVs between Different Climatic Zones

Climate is a key factor influencing surface cover, with climatic zones with different ESVs exhibiting significant spatial heterogeneity. Reference was made to Lin et al. [33] and their classification of 13 climatic types (Figure 6).

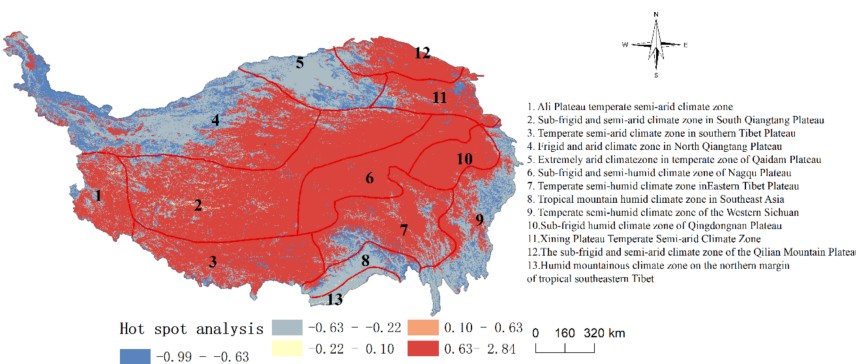

**Figure 6.** Ecosystem service values under different climate regions in the QTP.

After superimposing the figure above onto the QTP's ESV map, the areas with low ecosystem service value on the QTP are mainly concentrated in the southwest and northwest of the QTP. The low value area of ecosystem service value in the southwest of the Qinghai-Tibet Plateau is mainly distributed intropical mountain humid climatic zone in southeastern Tibet, mountain humid climatic zone (Ia) in the northern margins of southeastern Tibet, and a temperate humid climatic zone (HIIIa) in the western Sichuan Plateau. The northwest region is mainly distributed in the extremely arid temperate climatic zone associated with the Qaidam Plateau, and the cold arid climatic zone associated with the northern Qiangtang Plateau. The conditions of the arid climatic zones of the northwestern QTP are under the influence of their geographical locations. It is difficult for warm and humid air currents to reach the locations because they are surrounded by mountains and are far from the oceans. The resultant arid climates are not conducive for vegetation growth, resulting in widespread bare land. Conversely, the humid area in the southeast has excellent moisture and temperature conditions, with the predominant vegetation being subtropical evergreen broad-leaved forests. However, the overall ESV is low due to limited woodland area.

The sub-frigid semi-humid climatic zone of the Nagqu Plateau and the sub-frigid humid climatic zone of the southeastern QTP are areas with high ESVs. The two climatic zones are under the influence of weather features such as shear lines. Pasture growth in the area is excellent and the grassland is contiguously distributed, leading to the high ESVs. The temperate arid climatic zone of the Ali Plateau (which is a mountain–plateau border zone), the sub-frigid semi-arid climatic zone of the southern Qiangtang Plateau, and the sub-frigid semi-arid climatic zone of the Qilian Mountains plateau are all areas with alternating climates. The ESVs of the climatic zones are mainly influenced by the differences in humidity conditions and natural resource endowment between the north and south, and the east and west. Their ESVs lie between those of two states described previously.

### 3.4.3. ESV Characteristics at Different Altitudes

The QTP ESVs are controlled by the scale and combination of land use types, resulting in dissimilar characteristics at different altitudes. There is a critical point at 6000 m above sea level (ASL), beyond which there are vast changes in temperature and precipitation (Figure 7). The vegetation cover types become simplified, resulting in a substantial decline in the ESVs. In terms of the overall characteristics, the ESVs within the same altitudinal ranges decreased before increasing, with rising trends on the whole. Across the different altitudinal ranges, the ESVs of areas below 3000 m ASL exhibited fluctuating increases. These areas have dense population distributions and urbanization, resulting in large-scale land occupation and exploitation, all of which have led to the permanent destruction of already-damaged ecosystems.

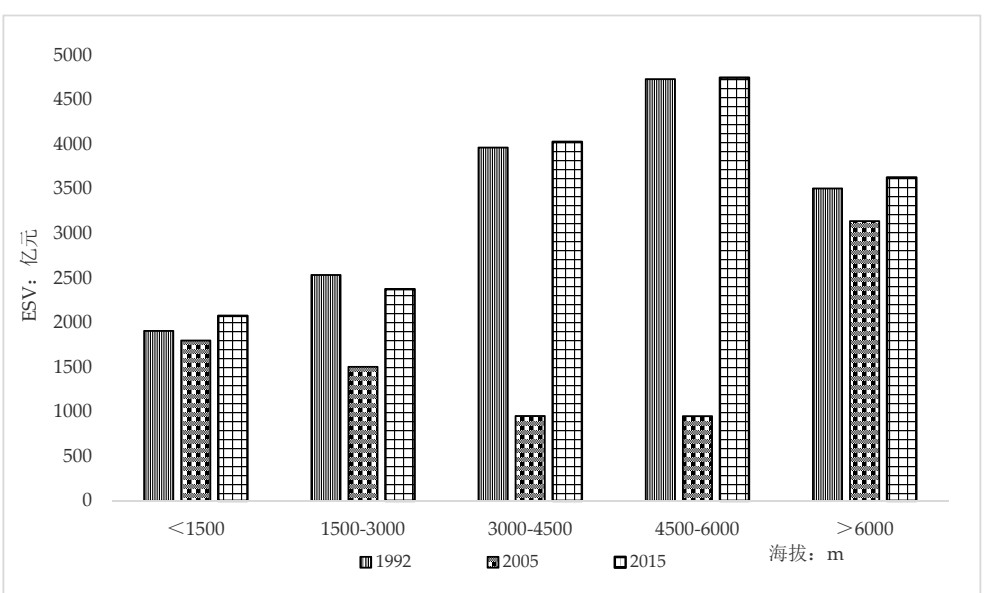

**Figure 7.** Ecosystem service value trends under different elevations on the Qinghai-Tibet Plateau.

In the 3000–6000 m ASL range, the ESVs declined sharply at first and then increased rapidly. The reason is that the altitudinal range hosts numerous human activities, including urbanization, intensive livestock grazing and rearing, and transportation infrastructure associated with the QTP's rapid development in recent years. After a dramatic decline in 1992–2005, the regional ecology of areas at the altitudinal range gradually improved and their ESVs were enhanced substantially, which is attributable to the enhancement of ecological conservation activities after 2005. The measures were achieved following the implementation of the Operation Green Shield program and real-time monitoring using remote sensing technologies, which provided full cover and facilitated the monitoring of all nature reserves. In addition, many illegal buildings were demolished and numerous activities with negative impacts on the ecological environment were banned.

Areas above 6000 m ASL are not suitable for human habitation. The main land use types are frozen soil, glacial snow, and alpine meadows, all of which have inherently low ESVs. The changes observed at the altitude are relatively mild because human activities are limited, with sporadic alpine grazing being the only disturbance, therefore, human activities have less impact on the ESVs.

*3.5. Simulation of ESVs under Future Land Use Change Scenarios*

The earlier analyses show that QTP land use change is a key factor driving ESVs, with both variables exhibiting a strong coupling consistency in the dimensions of time and space. To ensure that the research results better serve the future developmental timeframe, two future scenarios were evaluated based on actual policies formulated for QTP ecological conservation. The first scenario is natural development, which is the continuation of the land use change law of the QTP from 1992 to 2015; the second scenario placed emphasis on ecological protection. A comparative analysis was carried out on the effects of regulation and control on changes to both land use and ESVs, so as to provide a theoretical basis for the formulation of appropriate regional policies.

3.5.1. Analysis of Future Land Use Changes under Different Scenarios

The areas of deserts, shrubland, bare land, and woodland increased under the natural development scenario. Deserts had the largest increase, amounting to 54,914 km$^2$ (6.30%), followed by shrubland, bare land, and woodland, with the rates of change being 2.10%, 0.97%, and 0.60%, respectively. Farmland had the greatest decrease (−0.81%), followed by

glaciers, grassland, and urban land (Table 4). Overall, urban land, shrubland, and farmland had transformation rates above 50%, with urban land having the highest rate, at 71.71%, and becoming mainly grassland, farmland, and bare land. The rate of transformation of shrubland was 56.48%, mainly into grassland and bare land. Farmland mostly became grassland and woodland, with the amount transformed into grassland being as high as 5714 km$^2$. All three land use types with the high transfer rates were largely transformed into grassland. The rates of transformation into deserts, urban land, and shrubland were all higher than 70%, with deserts being the highest at 74.86%. Approximately 80.23% and 18% of areas that became deserts were transformed from grassland and bare land, respectively. The rates of transformation into urban land and shrubland were 70.36% and 70.07%, respectively. Urban land was mainly transformed from farmland and grassland, while shrubland was mainly transformed from woodland and grassland. The rates of transformation either into or from urban land and shrubland were relatively high. In the case of shrubland, the difference between the two directions of transformation was 3874 km$^2$, with the amount of land transformed into shrubland being able to offset the amount transformed from urban land.

**Table 4.** Ecosystem services and change rates of various land types in the Qinghai-Tibet Plateau in 2035.

| Land Use Type | Rate of Change of Land Use Type | | Ecosystem Service Value/ 10$^8$ yuan | | ESV Change | |
|---|---|---|---|---|---|---|
| | Natural Development Scenario | Ecological Protection Scenario | Natural Development Scenario | Ecological Protection Scenario | Natural Development Scenario | Ecological Protection Scenario |
| Farmland | −0.81% | 0.65% | 18.95 | 28.1 | −23.77% | 13.03% |
| Woodland | 0.60% | 0.17% | 1257.93 | 1161.11 | 12.04% | 13.31% |
| Grassland | −0.50% | −0.13% | 3164.28 | 3418.42 | −9.91% | 1.24% |
| Shrubland | 2.10% | 0.38% | 31.22 | 23.62 | 42.10% | −2.68% |
| Waterbody | −0.01% | 0.30% | 2459.9 | 2614.87 | −0.24% | 3.41% |
| Desert | 6.30% | 0.67% | 17.84 | 8.94 | 126.11% | 7.51% |
| Bare land | 0.97% | 0.05% | 15.43 | 13.08 | 19.43% | 6.05% |
| Glacier | −0.60% | −0.66% | 29.94 | 29.54 | −12.02% | −13.19% |
| Total | | | 6995.49 | 7297.68 | −2.88% | 1.32% |

Under the ecological conservation scenario, the magnitudes of the various land use changes in 2015–2035 were relatively mild, with the overall trends being positive. The exceptions were grasslands and glaciers, which showed slight decreases. The reduction in the area of glaciers was the largest, at −0.66%, while that of grassland was only, −0.13%. In comparison, the area of urban land increased the most, from 2369 to 3068 km$^2$ (1.48%). Under the ecological conservation scenario, shrubland had a transformation rate of 50%, at 5131 km$^2$ and 60.15%. The lands were mainly transformed into grassland and bare land, with grassland land being the main direction. The rates of transformation of land use into shrubland, urban land, and farmland were all above 50%. The rate for transfer into shrubland was the highest at 63.81%, with the main sources being woodland, grassland, and bare land. Both urban land and farmland mainly originated from grassland. Under the ecological conservation scenario, the simultaneous conversions among grassland, woodland, and shrubland were prominent.

Between the two future scenarios, land use structure change was balanced, and the expansion of deserts and bare land was under control. Due to the limited amount of bare land on the QTP and the objective under the ecological conservation scenario being "to develop while protecting and to protect while developing," it is necessary to consider population growth and developmental needs. Therefore, there would still be minor expansions in farmland and urban land. Glacial melting arising from global climate change would also occur.

### 3.5.2. ESV Trends under Different Scenarios

ESVs under the natural development and ecological conservation scenarios were 699.549 B and 729.768 B yuan, respectively. Compared with the situation in 2015, the natural development scenario led to a deterioration, with the ESVs decreasing by 2.88%. In contrast, the ESVs increased by 1.32% under the ecological conservation scenario, indicating an improvement in the overall ecological environment (Table 4).

Under the natural development scenario, farmland, grassland, waterbody, and glacier ESVs decreased by 23.77%, 9.91%, 0.24%, and 12.02%, respectively. Among them, the farmland ESVs declined the most due to the encroachment of urban land. Deserts had the greatest ESV increase, from 789 million to 1.784 B yuan, reflecting, to a certain extent, the proliferation of the land use type. The magnitude of change in ESVs under the ecological conservation scenario was small. Only shrubland and glaciers exhibited ESV reductions; however, they were not considerable (2.68% and 13.19%, respectively). The ESV reduction for glaciers was greater than that for shrub, with the decline associated with glaciers being related to global warming and population pressure. During the simulation of the ecological conservation scenario, many restrictions were placed on the transformation of deserts, to increase their transfer costs. Nevertheless, the land use type still had a significant increase in area, leading to an increase in the ecological values of deserts by 7.51%. However, this was still a vast improvement compared with the 126.11% increase under the natural development scenario (Figure 8).

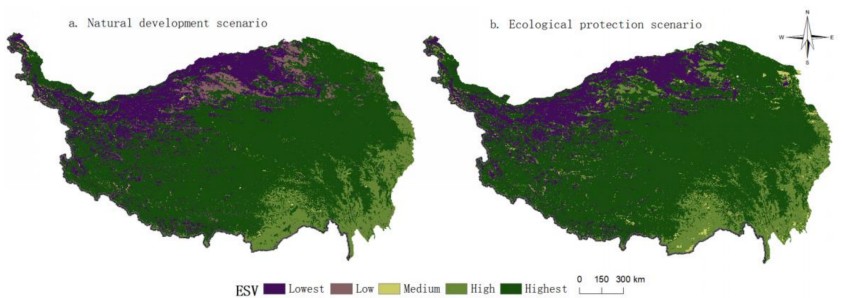

**Figure 8.** Spatial distribution of service value of the QTP ecosystem under different scenarios.

## 4. Discussion

This paper has four important findings on the relationship between land use change and ecosystem service value. Firstly, QTP land use changes were consistent with ESV trend, consistent with other studies, the trend of improvement was generally maintained [34], but it is attributable to changes in the properties of individual land use indicators arising from land use changes triggering a dynamic dependency effect with corresponding changes in the ESVs. Secondly, the changes in land use on the Qinghai-Tibet Plateau indicate that climate change and human activities are the two main causes of changes in land use structure, and the irreversible land use change caused by these two factors is the key factor that leads to the permanent damage of ecosystem value. Thirdly, eco-geographical area, climatic zone, and altitude were the key factors affecting QTP ESVs. Especially, areas below and above 6000 m presented dissimilar trends due to the different influential factors, such as interactions between land use structure and human activity limitations. The altitude factor of the Qinghai-Tibet Plateau affects the evolution of other elements through the control of climate and temperature. Fourth, enhancing the implementation of ecological and environmental protection policies on the current basis is conducive to improving the value of ecosystem services in the future, because policy intervention is undoubtedly a key measure to regulate ecological protection and human activities [35]. This further supports the efforts of China's environmental protection policies to enhance the value of ecosystem services.

The directions of transfer were dissimilar across the different land use types. In the past 23 years, the prominent changes were increases in urban land and deserts and

decreases in shrubland and bare land. For glaciers and waterbodies, there were linear and inverse relationships between their increases and decreases. Based on the research of Xu et al., the QTP's ESVs in 1995–2015 were mainly stable or rising [36]. Zhang et al. found that changes in the area of level-1 land use types amounted to less than 7%, and there was an overall improvement in land cover conditions [37]. The findings are consistent with the results of the present study. QTP land use changes showed a deteriorating trend with the expansion of urban land and bare land. However, China's regulatory and governance policies on ecological protection after 2005 greatly reversed the trend, resulting in the transformation of other land use types into grassland. Consequently, ES supply was improved regionally.

Climate change and human activities: these two reasons have overall and local effects on the land use of the Qinghai-Tibet Plateau, which affects the changes in the value of ecosystem services. Climate change has been particularly prominent in the region over the past 50 years, with the warming in the region being two-fold the global average over the same period. Such changes have led to glacial retreat, permafrost degradation, and avalanches, which have generally influenced the land use change in the region. The sustained melting of glaciers has led to a reduction in area by 2.84%. Separately, climate warming has triggered overall warming and humidification, changes in precipitation, and acceleration of the water cycle, with water vapor transported inland from the oceans and evaporation from humid inland areas increasing, and, in turn, increasing the amount of water vapor in the atmosphere [38]. Such trends have further accelerated the transformation of shrubland into woodland, bringing about an overall increase in ESVs. Bare land, shrubland, and glaciers exhibited declining ESVs to varying degrees. The conversion of glaciers to water bodies has increased the value equivalent of ecosystem services, but in terms of actual development, glacier melting has also brought other negative effects, such as the formation of barrier lakes, glacial lake outbursts, rainfall changes, natural landscape degradation, etc., as well as changes in grassland types, which have led to large fluctuations in grassland productivity, degradation, and desertification, and other negative effects, that will in turn affect the ecological service capabilities of animal husbandry, crop growth, and tourism development. Besides, human activities are certainly influencing and altering the fragile ecosystems in the QTP. The impact of human activities was primarily due to rapid population growth in the plateau as a result of economic activities such as tourism. The outcome was a rapid expansion of urban land, whose area increased by 96.92%. It is difficult to reverse such permanent changes. Land degradation caused by increased grazing is reversible. Transitional grazing in the early stages of the study played a key role in grassland degradation. However, after its restriction in the latter stages, the ecological environment improved significantly, leading to overall enhancements in the ESVs.

Simulations were carried out under two scenarios: natural development and ecological protection in this paper. Land use change was more balanced under the latter scenario, and the proliferation of deserts and bare land was inhibited to a certain extent. ESVs generally increased under the ecological protection scenario, resulting in enhanced sustainable development, indicating that China's environmental policies had a beneficial effect [39]. From 2000, the Chinese government and scholars began researching and managing the grassland–livestock balance challenge on the QTP. Overgrazing was reduced through the formulation and implementation of grazing prohibitions on the grassland and the grassland–livestock balance system. Other issues that were addressed included land desertification and surface soil erosion which were associated with excessive growth in the populations of wild animals that destroyed the high-altitude alpine grassland vegetation. Such actions have optimized the land use structure and ES capacity to a large extent. Under this basic law, the green development of the Qinghai-Tibet Plateau must consider differentiated regulation and dynamic adjustment. Differentiated regulation is mainly to establish land use methods that are coordinated with the dominant functions of natural ecological areas, and establish green energy levels such as national parks, natural reserves, and natural parks through large-scale protection and small-use models in large areas

of primitive habitats. Encourage the development of ecotourism rather than excessive grazing to increase the value of ecosystem services; an active population and industrial urbanization policy should be implemented in areas with a foundation for urbanization and relatively abundant water resources, and relocate people from high altitudes (greater than 5000 m), which is not suitable for human habitation. In terms of dynamic adjustment, it is necessary to seek the leading influencing factors in areas where ecological value increases and decreases, and to conduct dynamic observations on the scale of population urbanization, agriculture and animal husbandry, and the scale of wildlife growth, and scientifically and rationally adjust the balance between them. To build a friendly ecological protection and utilization pattern within the scope of carrying capacity.

## 5. Conclusions

To achieve the research aim (to reveal the relationship between land use change and ESVs in the Qinghai-Tibet Plateau), a comprehensive theoretical literature in land science, ecology, and economics is reviewed to establish a conceptual framework of land use transfer matrix and ESVs, and evaluated the change of ecosystem service value under the two scenarios in the future. This paper adopts the method of ecosystem function research in the Qinghai-Tibet Plateau by predecessors and carried out a confirmative case study on the changes of ecosystem service value on the Qinghai-Tibet Plateau from 1995 to 2015, which found that the data results calculated by the model method were in good agreement with previous studies, which proved that the method and case selection in this paper were appropriate and universal. It can provide reference for other related research. We summarise some key theoretical implications here.

It is observed that the impact of land use change on the ESVs was synchronous and positive in the Qinghai-Tibet Plateau, although the change of some indexes weakened the value, the land use structure and ESVs of the Qinghai-Tibet Plateau generally developed in a good direction. In the process, climate change and topography are the leading factors affecting land use and ESV changes, while human activities played an important regulatory role in local areas. Proactive ecological protection Policies could effectively reverse the QTP's Trend of land use deterioration and optimize its ESVs supply capacities. So, in order to alleviate the negative impact of climate change, it is necessary for the Qinghai-Tibet Plateau to always adhere to green economic policies such as agriculture, animal husbandry, and tourism, and joint efforts of global economies to maintain the integrity and authenticity of the Qinghai-Tibet Plateau ecosystem from irreversible damage. Active improvements should be made in controlling the amount of agriculture, animal husbandry, and livestock, controlling the disorderly development of urbanization, and advocating green and low-carbon tourism behavior.

However, although ecological protection policies need to be adhered to for a long time, this result cannot conceal the current policy crisis, and future evaluation and improvement of the rationality and effectiveness of policies are needed. In the future, China will still need to promote the protection of the Qinghai-Tibet Plateau through regional ecological public welfare forest policies, ecological transfer payments, and ecological compensation, explore new models for integrating farmers and herdsmen into the protection system is also needed. The current subsidy policy is based on the grassland area owned by the basic family unit and the unit compensation standard for ecological protection subsidies, as well as ethnic population subsidies, border residents' subsidies, etc., and it has also formed some negative effects, including long-term government subsidies. Development inertia, human-to-livestock conflicts caused by insufficient feed for returning grazing to the grassland, and the fencing economy, and the difficulty of changing the way of animal husbandry caused by the limitation of development capacity. Their existence has caused large-scale payment pressure from the central government and the development of Tibetan governments and members of society. There is an urgent need to jointly explore new ways of land use in the Qinghai-Tibet Plateau and use new development models to promote the development of the Qinghai-Tibet Plateau during the period of ecological civilization. How can we

stimulate the spontaneous motivation of local residents to optimize the land use structure and its ecosystem services? These still require a lot of scientific investigation and analysis, including comprehensive consideration of multiple dimensions such as education quality, employment skills, national concepts, industrial models, and protected area policies.

The present study had a key limitation with regard to methodology. It involved the combination of the equivalent factor per unit area algorithm with remote sensing image analyses to objectively analyze the patterns of change in land use in the QTP and ESVs, before carrying out preset simulations of future scenarios (with and without protection). It is unclear to what extent the simulated results can reflect the overall state. In future studies, the methodology could be optimized and improved by including an assessment of the qualities of different land cover types. In addition, the double threats of global change and human activities also persist in Central Asia, West Asia, Africa, and other regions [40]. The results achieved for the other regions were quite dissimilar to that of China due to the different governance models adopted. In the future, the relationships and the mechanisms governing them should be compared and analyzed in other ecosystems, which would enhance our understanding of the factors influencing land use structure and ES optimization under different socioeconomic conditions.

**Author Contributions:** Methodology, H.Y.; software, L.X.; investigation, X.Z., Q.L.; writing—original draft preparation, Y.Z.; writing—review and editing, H.Y.; supervision, H.Y. All authors have read and agreed to the published version of the manuscript.

**Funding:** This research was funded by the Joint Research Project by the Chinese Academy of Sciences (CAS) and the Sanjiangyuan National Park under the Qinghai Province People's Government (YHZX-2020-07), the 2nd Comprehensive Scientific Investigation and Research Project of the Qinghai–Tibet Plateau (2019QZKK0401), and Special Project for Type-A Strategic and Leading Technologies under the CAS (XDA20020301).

**Institutional Review Board Statement:** Not applicable.

**Informed Consent Statement:** Not applicable.

**Data Availability Statement:** The raw/processed data required to reproduce these findings cannot be shared at this time as the data also forms part of an ongoing study.

**Acknowledgments:** The authors thank all researchers involved in data acquisition, image registration, and urban lands delineation and verification. They were members and students in the land resources, remote sensing research department, Chinese Academy of Sciences.

**Conflicts of Interest:** The authors declare no conflict of interest.

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
