# Peer review of "Land Use-Driven Changes in Ecosystem Service Values and Simulation of Future Scenarios: A Case Study of the Qinghai–Tibet Plateau"

_sustainability, doi:10.3390/su13074079_

Round 1

Reviewer 1 Report

The study evaluates land use changes in the Qinghai–Tibet Plateau in the years 1992, 2005, and 2015 to determine their impacts on ecosystem service values (ESV). The main aims were to explore (I) land use changes in the case study, (ii) spatial ESV trends under land use change, and (iii) future ESV trends under natural evolution and ecological protection scenarios.

The study seems to be a valuable contribution in the ESV discipline based on a research in an attractive case study. The study is well structured and written.

I have several comments:

1) The land use change analysis is the most important part of the study. However, there is a lack of information/metadata about data used. Please provide more detailed description of the land use data type. What is an original source of the data? Why did you select this source of data with lack of open information? Why did not use Open Data source, e.g. remote sensing data or open data derived from remote sensing? Provide the rationale for choosing this land use data.

2) What is a classification nomenclature, description of land use data classes? What is e.g. a difference between desert and unused class? A class of bare land is mentioned in the paper. Where is this class included in the land use classification system? There is no description about the land use classes used in the paper.

3) There is no information or discussion about quality/accuracy of the land use data. It should be added a part in the paper discussing a reliability of the input data. What is an accuracy of the land use dataset used?

4) Some land use analysis and driving forces definitions are confusing. For example, l. 289-290: “There was also an obvious increase in desert area (17.31%) due to temperature, precipitation, and climate warming factors.” This sentence is in the opposite with other sentence: “Deserts and shrub land, with a total area of 5,093 km2, transformed predominantly into grassland, with discernible ecological improvements, including the increase of water storage capacity. (l. 313 – 315).  Check and describe better and more accurate the land use changes. The results about land use changes are sometimes confusing. Please clarify and check the results of the land use changes.

5) Based on results of the land use changes, driving forces are not exactly defined. Driving forces  are defined in vague way, e.g. a class desert: Is really the main driver of the changes of the desert “temperature” and “precipitation”? What theoretical system/concept of the driving forces has been used in the study?

6) Discussion and Conclusion: Instead of just repeating the findings, the discussion and conclusion should express an importance of the study.

7) From my point of view, the study has a strong descriptive character with a combination of standard research methods used. Could you explain and mention in the text: how this study contributes to the scientific community? What is an added value of this study? What is a new contribution from point of view methods?

8) Figure 1: to add „ m a.s.l. „ into legend of DMT. Why does Fig. 1 contain two maps? Could you add the most important cities into coloured map and remove the second one map?

Author Response

JAN. 19, 2021

Responses to Reviewer’s Comments

Dear Reviewer:

Thank you for reviewing our manuscript, “Land Use-Driven Changes in Ecosystem Service Values and Simulation of Future Scenarios: A Case Study of the Qinghai–Tibet Plateau” [Manuscript ID: sustainability-1058234]. As per your comments and suggestions, we have made significant revisions to our manuscript. Please find attached our point-by-point responses to your concerns. We sincerely hope that you find the responses satisfactory.

Sincerely yours,

The Authors

Response to Reviewer 1’s Comments

  1. The land use change analysis is the most important part of the study. However, there is a lack of information/metadata about data used. Please provide more detailed description of the land use data type. What is an original source of the data? Why did you select this source of data with lack of open information? Why did not use Open Data source, e.g. remote sensing data or open data derived from remote sensing? Provide the rationale for choosing this land use data.

The description of modify:

The use of land use Data mainly comes from national institute of science Data center of the Qinghai-Tibet plateau, (China's first by Scientific Data authentication Data storage center), the Data center is given priority to with the Qinghai-Tibet plateau and the surrounding areas of various kinds of Scientific Data, is China's only Scientific Data on the Tibetan plateau and surrounding areas with the most comprehensive category, the most authoritative Data center, Data center at the same time relying on the third pole environment (TPE) international plan for international cooperation. For use in this study, 1992, 2005 and 2015 of the Qinghai-Tibet plateau land use data is mainly the data center is based on 2015 ESA ESA GlobCover global land cover data, land use data by combining geographical resources of Chinese Academy of Sciences NLCD - China, Tsinghua university global land cover FROM GLC, NASA MODIS data - global land cover MCD12Q1 data, the university of Maryland global land cover UMD, the USGS land cover data IGBP DISCover, The conversion rules of LUC classification system of Qinghai-Tibet Plateau and other data classification systems were constructed, the confidence function of land cover classification and land type fusion rules were constructed, the land cover product fusion and correction were carried out, and the land use data of Qinghai-Tibet Plateau V1.0(1992, 2005, 2015 300m×300m grid, first-level classification) was completed. The details are as follows (see P5) :

2.2. Data Sources

Spatial data of the administrative regions, land use data, socioeconomic development statistics, and natural and geographic environment data were used as explanatory variables in the present study. The respective sources were as follows: (i) spatial data on administrative regions were acquired from the official website of the State Bureau of Surveying and Mapping; (ii) land use data at a resolution of 300 m × 300 m for three periods of 1992, 2005, and 2015[39], were obtained from the National Tibetan Plateau Data Center (http://data.tpdc.ac.cn/zh-hans/). The data were mainly based on the 2015 ESA GlobCover data. The conversion rules of the Qinghai-Tibet Plateau land cover classification system were constructed, the confidence function of land cover classification and land type fusion rules were constructed, and the land cover product fusion and modification were carried out. The production of this land use data has gone through five stages: remote sensing image data preparation, classification system establishment, interpretation mark establishment, classification interpretation stage, and accuracy inspection of interpretation results. The accuracy inspection mainly relies on high score images and field Empirical combination, so the Kappa value of the land use data is significantly higher than 90% ; (iii) socioeconomic development statistics (equivalent factors for value enhancement), including grain output per unit area, total crop income, and national grain output per unit area of the various QTP counties, were primarily obtained from the China Statistical Yearbook 2016, and statistical yearbooks of the respective provinces and cities; (iv) geomorphological data, including elevation, topography, and surface water characteristics (rivers and lakes), were downloaded from the Geospatial Data Clouds. After exploring relevant literature[40], seven explanatory variables were selected from the natural, socioeconomic, and transportation dimensions, including altitude, slope, population per unit area, night-time light data, and respective distances from rivers/lakes, roads, and the city center .

  1. What is a classification nomenclature, description of land use data classes? What is e.g. a difference between desert and unused class? A class of bare land is mentioned in the paper. Where is this class included in the land use classification system? There is no description about the land use classes used in the paper.

The description of modify:

The desert mentioned in the article refers to the special land covered with sand and basically no vegetation, excluding the sandy land in the tidal flat, while the bare land is the land with the surface layer of soil and basically no vegetation cover, or the surface layer is rock, gravel , Which covers an area of ≥70% of the land. There are significant differences in surface cover between the two types of land, but both of them belong to other land in the classification of current land use in China. The bare land and unused land appearing in the article are mainly due to the author's mistakes, who accidentally misinterpreted the bare land as unused land. The specific land use classification names and descriptions are as follows:

Three categories

Class

Meaning

Agricultural

land

Farmland

Land for growing crops, including cultivated land, newly developed land, reclaimed land, reclaimed land, and fallow land (including roving land and crop rotation land).

Grassland

Land that is dominated by vegetation, vegetation and vegetation

Forest

Land on which trees, bamboos and shrubs grow, and land on which mangroves grow along the coast, excluding land for afforestation trees inside residential areas, trees within the scope of land expropriated by railways and highways, and berm forests of rivers and ditches

Shrub

land

The vegetation types dominated by shrubs, and most of the building species are mesophytic and clustered shrub life forms

Construction

land

Construction

land

Land on which buildings and structures are built, land on which various objects are built

Unused

land

Desert

The surface of the land covered by sand, almost no vegetation cover, excluding the land in tidal flats

Bareland

The surface soil, basically no vegetation cover land; Or the surface of the rock, gravel, its coverage of more than 70% of the land

Waterbody

Rivers, lakes, canals, channels, reservoirs, ponds and their areas of control as well as hydraulic facilities, excluding sea areas and fish ponds dug on cultivated land.

Glacier

Land covered with snow and ice all year round

Based on expert suggestions, relevant content on the classification and description of land use types has been added to the article. The specific content is as follows (see P8):

3.1.1. Changes in land use structure

According to the comparison table of "Three Categories" in "Land Use Status Classification" and "Land Management Law of the People's Republic of China", the land cover classification system of the Qinghai-Tibet Plateau was constructed, and the land cover types in the Qinghai-Tibet Plateau were divided into nine categories: Farm land , Construction land, Desert, Bare land, Grassland, Forest, Shrub, Waterbody and Glacier. Local adjustments occurred in QTP land use structure over the 23-year study period, with the two prominent changes being increase in urban land and desert areas, and decrease in shrub land and bare land (Figure 3). The QTP was predominantly grassland in 1992–2015, with the land use type accounting for 67.91% of the total area. This was followed by bare land and woodland, accounting for 14.59% and 9.01% of the total area, respectively. The three land use types jointly comprised 91.51% of the total area, with no obvious change in their predominance over time. Driven by urbanization on the plateau and expansion of urban construction, the growth rate of urban land area reached 96.92%. There was also an obvious increase in desert area (17.31%) due to temperature, precipitation, and climate warming factors. Correspondingly, there were reductions in shrub land and bare land areas, at -4.41%and-9.42%, respectively. In addition, glacial melting decreased by 2.84%.

  1. There is no information or discussion about quality/accuracy of the land use data. It should be added a part in the paper discussing a reliability of the input data. What is an accuracy of the land use dataset used?

The description of modify:

The land use data of the Qinghai-Tibet Plateau used in this article was completed by Xu Erqi et al under the funding of the Pan-Third Pole Environment Study for a Green Silk Road-A CAS Strategic Priority A Program (XDA20000000). The data production process is mainly remote sensing image data. There are five stages: preparation, establishment of classification system, establishment of interpretation mark, stage of classification and interpretation, and accuracy inspection of interpretation results. During the production of land use data, 100% inspections were carried out at all stages, and spatial sampling inspections were performed on the acquired land use data to verify whether the land use data types are qualitatively accurate. The verification method mainly relies on high-resolution images and the field. The combination of empirical research shows that the Kappa value is significantly higher than 90% after the accuracy verification of the land use data, indicating that the accuracy of the land use data is relatively high.

A discussion on the accuracy of land use data has been added to the article in accordance with expert suggestions. The specific content is as follows (see P5):

2.2. Data Sources

Spatial data of the administrative regions, land use data, socioeconomic development statistics, and natural and geographic environment data were used as explanatory variables in the present study. The respective sources were as follows: (i) spatial data on administrative regions were acquired from the official website of the State Bureau of Surveying and Mapping; (ii) land use data at a resolution of 300 m × 300 m for three periods of 1992, 2005, and 2015[39], were obtained from the National Tibetan Plateau Data Center (http://data.tpdc.ac.cn/zh-hans/). The data were mainly based on the 2015 ESA GlobCover data. The conversion rules of the Qinghai-Tibet Plateau land cover classification system were constructed, the confidence function of land cover classification and land type fusion rules were constructed, and the land cover product fusion and modification were carried out. The production of this land use data has gone through five stages: remote sensing image data preparation, classification system establishment, interpretation mark establishment, classification interpretation stage, and accuracy inspection of interpretation results. The accuracy inspection mainly relies on high score images and field Empirical combination, so the Kappa value of the land use data is significantly higher than 90% ; (iii) socioeconomic development statistics (equivalent factors for value enhancement), including grain output per unit area, total crop income, and national grain output per unit area of the various QTP counties, were primarily obtained from the China Statistical Yearbook 2016, and statistical yearbooks of the respective provinces and cities; (iv) geomorphological data, including elevation, topography, and surface water characteristics (rivers and lakes), were downloaded from the Geospatial Data Clouds. After exploring relevant literature[40], seven explanatory variables were selected from the natural, socioeconomic, and transportation dimensions, including altitude, slope, population per unit area, night-time light data, and respective distances from rivers/lakes, roads, and the city center .

  1. Some land use analysis and driving forces definitions are confusing. For example, l. 289-290: “There was also an obvious increase in desert area (17.31%) due to temperature, precipitation, and climate warming factors.” This sentence is in the opposite with other sentence: “Deserts and shrub land, with a total area of 5,093 km2, transformed predominantly into grassland, with discernible ecological improvements, including the increase of water storage capacity. (l. 313 – 315).  Check and describe better and more accurate the land use changes. The results about land use changes are sometimes confusing. Please clarify and check the results of the land use changes.

The description of modify:

Regarding the contradiction between the two sentences pointed out by experts, ("There was also an obvious increase in desert area (17.31%) due to temperature, precipitation, and climate warming factors." (l. 289-290) and "Deserts and shrub land, with a total area of 5,093 km2, transformed predominantly into grassland, with discernible ecological improvements, including the increase of water storage capacity. (l.313-315)), the author has rechecked the data and revised the content of the article. The author is in the text What I want to express is that although there was a large-scale transfer of grassland to desert during 1992-2015, deserts were also transferred to grassland at the same time, and the area of desert to grassland transferred out of the grassland accounted for a prominent proportion of 11.25%. Therefore, the transfer of desert to grassland has a certain effect on reducing grassland area and grassland desertification. The specific content is as follows (see P9-10):

From 1992 to 2015, the direction of transfer of the QTP’s land use types occurred in the form two major structural changes, namely, toward ecological improvement and expansion of urban land use. Urban, desert, and shrub land accounted for relatively high proportions of the transformed areas. Land uses that were transformed included farmland, grassland, and waterbody, accounting for 93.09% of the total amount of transformed urban land (Table 2). There is a significant mutual conversion phenomenon between grassland and desert. About 5093km2 of desert area is converted into grassland, and the amount of grassland transferred to desert is 6937km2. Although the conversion between the two types of land is mainly from grassland to desert, desert to grassland To some extent, the transfer of grassland has alleviated the lack of grassland area. Shrub land was mainly transformed into woodland, accounting for 86.51% of the total area transformed. Glacier transformation into waterbody was also widespread.

  1. Based on results of the land use changes, driving forces are not exactly defined. Driving forces are defined in vague way, e.g. a class desert: Is really the main driver of the changes of the desert “temperature” and “precipitation”? What theoretical system/concept of the driving forces has been used in the study?

The description of modify:

Land is the carrier of human natural production and social economic reproduction. Under the intervention of human activities, the natural, social, economic, and ecological functions of land can be realized through the exchange of material, energy and information between land and the environment. The intervention of human social and economic activities promotes the creation of social productivity while renewing natural productivity. The land use system is a natural-economy-society-ecological complex system, showing the complexity of structure and function. The suitability factors involved in the simulation of land use in this study are the driving factors affecting land use change. When selecting suitability factors (driving factors for land use change), the author adopts empirical model summaries and a large number of empirical data. Statistical analysis methods. Due to the vast area of the Qinghai-Tibet Plateau as the study area, the regional differences in the driving forces of land use change are relatively obvious. Therefore, the author has selected the elevation, slope, distance from the natural and socioeconomic dimensions from a macro perspective and comprehensively based on relevant results. Seven factors such as distance between rivers and lakes, population per unit area, night light data, distance from roads, and distance from city centers are used as suitability factors affecting land use changes. For the Qinghai-Tibet Plateau, the fragile natural environment is the basis of land use changes, and human activities are the important drivers of land use changes.

  1. Discussion and Conclusion: Instead of just repeating the findings, the discussion and conclusion should express an importance of the study.

The description of modify:

(1) The changes in land use on the Qinghai-Tibet Plateau indicate that climate change and human activities are the two main causes of changes in land use structure. The QTP’s land use structure was relatively stable in 1992–2015. The predominant land use was grassland, and the supplementary land uses were woodland and bare land. The directions of transfer were dissimilar across the different land use types. In the past 23 years, the prominent changes were increases in urban land and deserts, and decreases in shrub land and bare land. For glaciers and waterbodies, there were linear and inverse relationships between their increases and decreases. The resultant ESVs improved in general. These two reasons have overall and local effects on the land use of the Qinghai-Tibet Plateau, which affects the changes in the value of ecosystem services. Grassland, water bodies and woodlands are stable suppliers of the value of the Qinghai-Tibet Plateau's ecological services. Grassland, waterbodies, and woodland were stable ES suppliers on the QTP. Bare land, shrub land, and glaciers exhibited declining ESVs to varying degrees. Although from a numerical point of view, the conversion of glaciers to water bodies has increased the value equivalent of ecosystem services, but in terms of actual development, glacier melting has also brought other negative effects, such as the formation of barrier lakes, glacial lake outbursts, and rainfall changes, natural landscape degradation, etc., as well as changes in grassland types, have led to large fluctuations in grassland productivity, degradation and desertification, and other negative effects, which in turn will affect the ecological service capabilities of animal husbandry, crop growth, and tourism development. Obviously, the Qinghai-Tibet Plateau is an early warning zone for global climate change. As the source of many rivers in inland China and Southeast Asia, the Qinghai-Tibet Plateau will also have a strong indirect effect on these regions and become a trigger and amplifier of natural environmental changes in other regions. To alleviate the negative impact of climate change, it is necessary not only for the Qinghai-Tibet Plateau to always adhere to green economic policies such as agriculture, animal husbandry and tourism, but also for the joint efforts of global economies, long-term attention and continuous adherence to the development of a series of natural ecological protection policies in the Qinghai-Tibet Plateau. Active improvements should be made in controlling the amount of agriculture, animal husbandry and livestock, controlling the disorderly development of urbanization, and advocating green and low-carbon tourism behavior.

(2) Qinghai-Tibet Plateau has a huge area, and Eco-geographical area, climatic zone, and altitude were the key factors affecting QTP ES. From the dimension of eco-geographical area, areas with increased ESVs were mostly concentrated in the Altun Mountains–Qaidam Basin belt. Conversely, areas with decreased ESVs were distributed in two centers: the southern part of the Kunlun Mountains–Altun Mountains and the eastern part of the southern Tibet valley. There was spatial heterogeneity based on a climatic zone perspective. The arid climatic zone in northwestern QTP and the humid climatic zone in southeastern QTP had lower ESVs, whereas the sub-frigid semi-humid climatic zone (HIVb) of the Nagqu Plateau and the sub-frigid humid climatic zone of the southeastern QTP (HIVa) had higher ESVs. In addition, areas below and above 6,000 m presented dissimilar trends due to the different influential factors, such as interactions between land use structure and human activity limitations. The altitude factor of the Qinghai-Tibet Plateau affects the evolution of other elements through the control of climate and temperature. Under this basic law, the green development of the Qinghai-Tibet Plateau must consider differentiated regulation and dynamic adjustment. Differentiated regulation is mainly to establish land use methods that are coordinated with the dominant functions of natural ecological areas, and establish green energy levels such as national parks, natural reserves, and natural parks through large-scale protection and small-use models in large areas of primitive habitats. Encourage the development of eco-tourism rather than excessive grazing to increase the value of ecosystem services; an active population and industrial urbanization policy should be implemented in areas with a foundation for urbanization and relatively abundant water resources, and relocate people from high altitude (greater than 5000m), which is not suitable for human habitation. In terms of dynamic adjustment, it is necessary to seek the leading influencing factors in areas where ecological value increases and decreases, and to conduct dynamic observations on the scale of population urbanization, agriculture and animal husbandry, and the scale of wildlife growth, and scientifically and rationally adjust the balance between them. To build a friendly ecological protection and utilization pattern within the scope of carrying capacity.

(3) Policy intervention is undoubtedly a key measure to regulate ecological protection and human activities. Simulations were carried out under two scenarios: natural development and ecological protection. Land use change was more balanced under the latter scenario, and the proliferation of deserts and bare land was inhibited to a certain extent. There were ESV losses under the natural development scenario, with the spread of deserts and bare land being the major factors responsible for the trends. ESVs generally increased under the ecological protection scenario, resulting in enhanced sustainable development, indicating that China’s environmental policies had a beneficial effect on ESs. However, this result cannot conceal the hidden crisis. Although in the future, China will still need to promote the protection of the Qinghai-Tibet Plateau through regional ecological public welfare forest policies, ecological transfer payments, and ecological compensation, explore new models for integrating farmers and herdsmen into the protection system is also needed. The current subsidy policy is based on the grassland area owned by the basic family unit and the unit compensation standard for ecological protection subsidies, as well as ethnic population subsidies, border residents subsidies, etc., and it has also formed some negative effects, including long-term government subsidies. Development inertia, human-to-livestock conflicts caused by insufficient feed for returning grazing to the grassland, and the fencing economy, and the difficulty of changing the way of animal husbandry caused by the limitation of development capacity. Their existence has caused large-scale payment pressure from the central government and the development of Tibetan governments and members of society. There is an urgent need to jointly explore new ways of land use in the Qinghai-Tibet Plateau and use new development models to promote the development of the Qinghai-Tibet Plateau during the period of ecological civilization. How can we stimulate the spontaneous motivation of local residents to optimize the land use structure and its ecosystem services? These still require a lot of scientific investigation and analysis, including comprehensive consideration of multiple dimensions such as education quality, employment skills, national concepts, industrial models, and protected area policies.

  1. From my point of view, the study has a strong descriptive character with a combination of standard research methods used. Could you explain and mention in the text: how this study contributes to the scientific community? What is an added value of this study? What is a new contribution from point of view methods?

The description of modify:

As Asian inland plateau, the Qinghai-Tibet plateau is the largest, the world's highest plateau, which is the most characteristic of a relatively independent natural ecological region, long-term human agriculture and urbanization affects the land use change and the Qinghai-Tibet plateau ecosystem services, northern Tibet from grazing line in nearly 40 years in northern Qiangtang no man's land, the original true to nature and biodiversity conservation has brought the huge pressure. The change of land use structure in the Qinghai-Tibet Plateau directly affects the supply of ecosystem service value, and its development and evolution play a key role in the construction of ecological security barrier in China and even the sustainable development in Southeast Asia. In the 1970s, China launched its first large-scale scientific expedition, involving more than 2,000 scientific researchers, with the focus on getting a basic geographical overview of the Qinghai-Tibet Plateau. This work continued until the 1990s, and basically formed a preliminary understanding of the Qinghai-Tibet Plateau. In the 21st century, China's economic and social growth model began to change. The sustainable development policy, especially the implementation of the main functional zones, focused on the construction of ecological civilization and green development, drew more attention to the role of the Qinghai-Tibet Plateau as an ecological security barrier. In 2017, the Chinese leader Xi jinping instructions and sent a letter to encourage scientists in China, with international counterparts together to strengthen to the Qinghai-Tibet plateau ecological security barrier system construction and the research of green development path, thus opens the second scientific research work of the Qinghai-Tibet plateau, China plans to invest 4 billion yuan, the work will focus on five years in all regions of the Qinghai-Tibet plateau, and surrounding countries and the scientific research, aimed at the first time research situation, on the basis of a clear focus of the 21st century after the "change" to a new survey of Qinghai-Tibet plateau, and strengthen the deepening research, Assessment in China in recent years on the Qinghai-Tibet plateau region development pattern, farming and animal husbandry development policy, ecological compensation policy, urbanization, etc., the area for the future development of new pattern and provide scientific conclusion support policy optimization, including natural protected area system construction and coordination pattern of human activity is one of the key direction. This paper is carried out under the background of this research, the practical and theoretical contribution is mainly manifested in the following several aspects:

(1) based on the latest aging data of scientific research, Chinese Academy of Sciences, a relatively long time series over the past 20 years were analyzed in the Qinghai-Tibet plateau under the influence of human activities, land use structure change and its trend, from on the whole, it is concluded that the characteristics of human activities on land use change basic understanding;

(2) based on the integrated approach of land use change and ecosystem service value, the trend prediction of natural development scenarios and ecological protection scenarios was carried out, and the influence laws of China's natural ecological protection policies on the development of land use and ecosystem services in the Qinghai-Tibet Plateau region were compared and analyzed.

(3) the method of theoretical contributions, this paper constructed the ecosystem services value of land use change prediction method, the CA - Markov model used a combination of the CA model to simulate the complex system space change ability and excellent properties of Markov process, more than a simple Markov model and CA model has better ability to simulate the evolution of complex adaptive system, the simulation also shows high accuracy when the prediction of land use.

  1. Figure 1: to add „ m a.s.l. „ into legend of DMT. Why does Fig. 1 contain two maps? Could you add the most important cities into coloured map and remove the second one map?

The description of modify:

The author has revised Fig 1 based on expert suggestions. The specific content is as follows (see P4).

Reviewer 2 Report

I have added an annotated paper will all the revision required. I consider that the paper should be reconsidered after major revision. Many aspects from conceptual background to actual description of the future changes (including description of the scenarios) should be much better explained. 

The conclusion part should be rewritten and so many other parts. The methods parts should also be rewritten. 

Author Response

JAN. 19, 2021

Responses to Reviewer’s Comments

Dear Reviewer:

Thank you for reviewing our manuscript, “Land Use-Driven Changes in Ecosystem Service Values and Simulation of Future Scenarios: A Case Study of the Qinghai–Tibet Plateau” [Manuscript ID: sustainability-1058234]. As per your comments and suggestions, we have made significant revisions to our manuscript. Please find attached our point-by-point responses to your concerns. We sincerely hope that you find the responses satisfactory.

Sincerely yours,

The Authors

Response to Reviewer 2’s Comments

  1. why is transportation outside socie-economic dimension?

The description of modify:

The author has revised Fig 1 based on expert suggestions. The specific content is as follows (see P4):

According to expert suggestions and a further understanding of the meaning of socioeconomic factors, socioeconomic factors refer to the economy, culture, and various social phenomena related to it, such as population, ethnicity, religion, agriculture, industry, Transportation, commerce, city, technology, etc. Therefore, the author decided to incorporate the transportation factor into the socio-economic factors and conduct research from the two dimensions of nature and social economy.

  1. The ecosystems are complex and diverse in many places.....are they different from other ecosystems? in what way? please cite or bring arguments.

The description of modify:

China has a vast territory and vast territory, which makes its ecosystem also present regional and complex characteristics. In order to effectively reflect the difference of ecosystem service functions in time and space, Xie Gaodi et al. based on the ecosystem service value equivalent factor algorithm, based on various literature surveys and biomass space-time distribution data, through the value of ecosystem services The equivalent factor table is revised and supplemented, and the improved ecosystem service value equivalent factor algorithm is collectively referred to as the improved equivalent factor algorithm.

Improved equivalent factor algorithm:

One standard unit ecosystem service value equivalent factor (hereinafter referred to as the standard equivalent) refers to the economic value of the annual natural grain output of 1 hm2 of the national average output of farmland. This equivalent is used as a reference and combined with expert knowledge to determine other ecosystem services. Equivalent factor, its function is to characterize and quantify the potential contribution of different types of ecosystems to ecological service functions. In practical applications, especially on the regional scale, it is difficult to completely eliminate the interference of human factors to accurately measure the economic value of the food output that the farmland ecosystem can provide under the natural conditions. This study refers to the treatment method of Xie Gaodi, etc., and takes the net profit of food production per unit area of ​​farmland ecosystem as the value of ecosystem services as a standard equivalent factor. The value of grain output in farmland ecosystems is mainly calculated based on the three main grain products of rice, wheat and corn. The calculation formula is as follows:

In the formula: D represents the value of ecosystem services of a standard equivalent factor (yuan/hm2); Sr, Sw, and Sc represent the percentage (%) of the sown area of rice, wheat, and corn in the total sown area of the three crops in a certain year ; Fr, Fw and Fc respectively represent the average net profit per unit area of rice, wheat and corn in a certain year (yuan/hm2).

  1. I found difficult to understand what means "Unused land". Can you explain the? Is there no ecosystem?

The description of modify:

The bare land and unused land appearing in the article are mainly due to the author's mistakes, who accidentally misinterpreted the bare land as unused land.

Bareland:The surface soil, basically no vegetation cover land; Or the surface of the rock, gravel, its coverage of more than 70% of the land.

  1. did you tested the trend? how can you see that it is a decreasing trend?

The description of modify:

From 1992 to 2015, the low-value area in the northern part of the Qinghai-Tibet Plateau showed an overall shrinking trend. Although during 1992-2005, due to the intensification of human activities and the delay of effective protection measures, the northern low-value area showed an increase in area. From 2005 to 2015, the state implemented a series of effective and localized protection policies, regulations, and measures for the Qinghai-Tibet Plateau. The growth trend of the low-value areas in the north was curbed and the area showed a trend of reduction. The decrease in the area of low-value areas in the northern part of the Qinghai-Tibet Plateau confirms to a certain extent the effectiveness of a series of protection measures by the state and local governments.

Round 2

Reviewer 1 Report

I would like to thank to the authors for a significant improvement of the paper.

However, there are still points, that should be reflected.

  1. l. 327 - 331 "There is a significant mutual conversion phenomenon between grassland and desert. About 5093km2 of desert area is converted into grassland, and the amount of grassland transferred to desert is 6937km2. Although the conversion between the two types of land is mainly from grassland to desert, desert to grassland To some extent, the transfer of grassland has alleviated the lack of grassland area."

This sentence is not understandable. Question: Are these processes related to the present time? Why you are using “present” time? In what year occurred, e.g. “About 5093km2 of desert area is converted into grassland”? Please clarify it and formulate it in better way. 

  1. The Discussion and Conclusions are not well structured and formulated. These parts are very important (the most important) parts of the paper. They should conclude and discuss the results and methods used and scientifically compare them with similar studies. They should be clearly formulated.

The Discussion is weak with descriptive character. Following recommendations:

Authors should discuss the results and how they can be interpreted in perspective of previous studies and of the working hypotheses. The findings, methods and their implications should be discussed in the broadest context possible and limitations of the work/methods highlighted. Future research directions should also be mentioned. Here you must respond to what the results mean. You need to make the Discussion corresponding to the Results, but do not reiterate the results.

Conclusions:

The conclusion creates a bigger picture of your research work that helps readers view the subject and results of your study as a whole. This section shows how the work advances the field from the present state of knowledge. You should provide a clear scientific justification for your work in this section. A short but clear form is recommended.

There are missing in the Discussion and Conclusions reflections for:  What is an added value of this study? What is a new contribution from point of view methods?

The current formulation of Discussion and Conclusions is not corresponding. It is a pity because the paper brings very interesting results and findings.

Author Response

Response to Reviewer 1’s Comments

  1. 327 - 331 "There is a significant mutual conversion phenomenon between grassland and desert. About 5093km2 of desert area is converted into grassland, and the amount of grassland transferred to desert is 6937km2. Although the conversion between the two types of land is mainly from grassland to desert, desert to grassland .To some extent, the transfer of grassland has alleviated the lack of grassland area."

This sentence is not understandable. Question: Are these processes related to the present time? Why you are using “present” time? In what year occurred, e.g. “About 5093km2 of desert area is converted into grassland”? Please clarify it and formulate it in better way.

The description of modify:

This sentence is not clear and does not distinguish between different areas of change. We have corrected and modified this. Please see the details P10(338-344).

We also searched and modified other confusing places in the paper.

2.The Discussion and Conclusions are not well structured and formulated. These parts are very important (the most important) parts of the paper. They should conclude and discuss the results and methods used and scientifically compare them with similar studies. They should be clearly formulated.

The description of modify:

We follow the suggestions of reviewers, and we have revised our discussions and conclusions in detail.  Please see the details P16-18.

Many thanks to reviewers for their suggestions, which greatly improved the quality of this paper. If anything wrong with the revision, please feel free to let us know and we will make further modifications.

Sincerely,

Authors

Reviewer 2 Report

I consider that the author din not improve the paper after the revision.

Still there are a lot of uncertainties and unanswered questions, like for e.g the difference between the two scenarios: "natural development scenarios" and "ecological protection scenarios"? The author is not providing any explanation on the meaning of such scenarios. Or on the phrase "Consequently, the evolutionary direction of ES values (ESVs) has become diverse" which does not make any sense and it is not explained.

Contradictory statements like for e.g. "Such impacts have increasingly become significant, especially in the Qinghai–Tibet Plateau (QTP), an ecologically fragile region....with "According to the results, the QTP land-use structure is relatively stable" are still present in the article.

Other aspects remains even after revision, including important aspects regarding the content of the paper, the question 2. The ecosystems are complex and diverse in many places.....are they different from other ecosystems? in what way? please cite or bring arguments. For this question no real answer has been given.

The same is for question 4. did you tested the trend? how can you see that it is a decreasing trend? No arguments and no testing of the trend being provided. 

Author Response

Edit Description

FEB. 5, 2021

Responses to Reviewers’ Comments

Dear Reviewers:

Thank you for reviewing our manuscript, “Land Use-Driven Changes in Ecosystem Service Values and Simulation of Future Scenarios: A Case Study of the Qinghai–Tibet Plateau” [Manuscript ID: sustainability-1058234]. As per your comments and suggestions, we have made significant revisions to our manuscript. Please find attached our point-by-point responses to your concerns. We sincerely hope that you find the responses satisfactory.

Sincerely yours,

The Authors

Response to Reviewer’s Comments

  1. I consider that the author din not improve the paper after the revision.

The description of modify:

In the last draft, we made detailed modifications to the opinions raised by the experts. However, due to the writing style problems in both Chinese and English, there are weak links in the general situation presentation, research discussion and conclusion, etc. This time, we made detailed modifications to these aspects.

  1. Still there are a lot of uncertainties and unanswered questions, like for e.g the difference between the two scenarios: "natural development scenarios" and "ecological protection scenarios"? The author is not providing any explanation on the meaning of such scenarios. Or on the phrase "Consequently, the evolutionary direction of ES values (ESVs) has become diverse" which does not make any sense and it is not explained.

The description of modify:

We analyzed and defined the two scenarios in detail. In the paper, the natural development scenario is constructed based on the assumption that the rate of land use conversion in the historical period remains unchanged [46], and the influence trend of various suitability factors on land use change will continue to be consistent with the current situation. Under the natural development scenario, the rules of transfer between various land use types in 1992–2015 were adopted as the baseline scenario for forecasting future land use change. Under the ecological protection scenario, areas within the QTP’s main functional regions that are prohibited from development were integrated with all the nature reserves, national parks, geological parks, and scenic spots being demarcated as zones in which development is restricted. These were combined with urban land and agricultural land expansion to comprehensively study and evaluate future development scenarios. The natural development scenario can be regarded as the benchmark scenario for future land use changes in the Qinghai-Tibet Plateau [47]. The ecological protection scenario is based on the natural development scenario, giving priority to the protection of basic farmland, ecological protection areas, etc. [48], and relaxing restrictions on urban development on the premise of respecting the natural ecosystem and reasonable environmental carrying capacity. This scene follows the principles of sustainable development and emphasizes the coordinated development of protection and development [49]. The ecological protection scenario is a comprehensive and sustainable future land use change scenario and an ideal scenario for future land use changes on the Qinghai-Tibet Plateau.

  1. Contradictory statements like for e.g. "Such impacts have increasingly become significant, especially in the Qinghai–Tibet Plateau (QTP), an ecologically fragile region....with "According to the results, the QTP land-use structure is relatively stable" are still present in the article.

The description of modify:

Here is a real error, in English is not easy to understand. The original intention is the same on the whole, but in the local show some differences. The relevant parts are modified in this paper.

4.Other aspects remains even after revision, including important aspects regarding the content of the paper, the question 2. The ecosystems are complex and diverse in many places.....are they different from other ecosystems? in what way? please cite or bring arguments. For this question no real answer has been given.

The description of modify:

The article did not take into account the needs of different readers, and assumed that everyone had a good understanding of the basic situation of China and the Qinghai-Tibet Plateau, so it did not go into detail. The basic information of this part has been added in this revision. Please See P5 (212-218).

  1. The same is for question 4. did you tested the trend? how can you see that it is a decreasing trend? No arguments and no testing of the trend being provided. 

The description of modify:

In the revision, the accuracy verification of scenario prediction is improved. Please see P8 (291-306).

2.5.6 Verifying accuracy of the simulated results

CROSSTAB, the Kappa coefficient verification method provided in IDRISI, was used to verify the simulation results and ensure their accuracy and reliability. This study uses the 2005 land use data as a benchmark and uses the CA-Markov model to simulate the distribution of land use in 2015. In order to measure the consistency between the simulation results and the actual land use data, this study uses the Kappa value to evaluate the reliability of the model. The accuracy verification shows that the Kappa value is 0.9216, and the Kappa value>0.85, indicating that the model has high simulation accuracy, meets the research needs, and can be used to predict future land use patterns. Protecting the ecosystems in the QTP is a major project of the Chinese government. The government has established 155 nature reserves at various administrative levels and in different stages of succession, accounting for 31.6% of the plateau’s total area. Since the 1990s, China has successively implemented numerous major ecological projects, including the ecological protection and development of Sanjiangyuan; protection and development of ecological security barriers in Tibet; ecological protection and restoration of the Qilian Mountains and its rivers, forests, farmland, and lakes; comprehensive management of the Qinghai Lake basin; and formulation of the Regional Plan for Ecological Development and Environmental Protection of the QTP (2011–2030) to strengthen ecological protection of the QTP.

And discussed the existing deficiencies, . Please see P18 (660-671).

The present study had a key limitation with regard to methodology. It involved the combination of the equivalent factor per unit area algorithm with remote sensing image analyses to objectively analyze the patterns of change in land use in the QTP and ESVs, before carrying out preset simulations of future scenarios (with and without protection). It is unclear to what extent the simulated results can reflect the overall state. In future studies, the methodology could be optimized and improved by including an assessment of the qualities of different land cover types. In addition, the double threats of global change and human activities also persist in Central Asia, West Asia, Africa, and other regions[65-67]. The results achieved for the other regions were quite dissimilar to that of China due to the different governance models adopted. In the future, the relationships and the mechanisms governing them should be compared and analyzed in other ecosystems, which would enhance our understanding the factors influencing land use structure and ES optimization under different socioeconomic conditions.

Many thanks to reviewers for their suggestions, which greatly improved the quality of this paper. If anything wrong with the revision, please feel free to let us know and we will make further modifications.

Sincerely,

Authors
